**Data Availability Statement:** Some relevant data has been provided via the manuscript and its Supporting Information files. Other data has been

# Highly sensitive molecular assay based on Identical Multi-Repeat Sequence (IMRS) algorithm for the detection of *Trichomonas vaginalis* infection

Clement Shiluli[1], Shwetha Kamath[2], Bernard N. Kanoi[1], Racheal Kimani[1], Bernard Oduor[3], Hussein M. Abkallo[3], Michael Maina[1], Harrison Waweru[1], Moses Kamita[1], Nicole Pamme[4], Joshua Dupaty[5], Catherine M. Klapperich[5], Srinivasa Raju Lolabattu[2]*, Jesse Gitaka[1]*

1 Centre for Research in Infectious Diseases, College of Graduate Studies and Research, Mount Kenya University, Thika, Kenya, 2 Division of Research and Development, Jigsaw Bio Solutions Private Limited, Bangalore, India, 3 Animal and Human Health Program, International Livestock Research Institute, Nairobi, Kenya, 4 Department of Materials and Environmental Chemistry, Stockholm University, Stockholm, Sweden, 5 Department of Biomedical Engineering, Boston University, Boston, Massachusetts, United States of America

* raju@jigsawbio.com (SRL); jgitaka@mku.ac.ke (JG)

## Abstract

### Introduction

Annually, approximately 174 million people globally are affected by *Trichomonas vaginalis* (*T. vaginalis*) infection. Half of these infections occur in resource-limited regions. Untreated *T. vaginalis* infections are associated with complications such as pelvic inflammatory disease and adverse pregnancy outcomes mostly seen in women. In resource-limited regions, the World Health Organization (WHO) advocates for syndromic case management. However, this can lead to unnecessary treatment. Accurate diagnosis of *T. vaginalis* is required for effective and prompt treatment. Molecular tests such as Polymerase Chain Reaction (PCR) have the advantage of having a short turn-around time and allow the use of non-invasive specimens such as urine and vaginal swabs. However, these diagnostic techniques have numerous disadvantages such as high infrastructure costs, false negative and positive results, and interstrain variation among others. This study aimed to evaluate the use of identical multi-repeat sequences (IMRS) as amplification primers for developing ultrasensitive diagnostic for *T. vaginalis*.

### Methods

We used genome-mining approaches based on identical multi-repeat sequences (IMRS) algorithm to identify sequences distributed on the *T. vaginalis* genome to design a primer pair that targets a total of 69 repeat sequences. Genomic *T. vaginalis* DNA was diluted from $5.8 \times 10^2$ to $5.8 \times 10^{-4}$ genome copies/$\mu$l and used as a template in the IMRS-based amplification assay. For performance comparison, 18S rRNA PCR assay was employed.

deposited in a repository. All raw data has been uploaded on Zenodo https://doi.org/10.5281/zenodo.14589291. All supplementary data has been uploaded on Zenodo https://doi.org/10.5281/zenodo.11176058.

**Funding:** This research was supported by the Royal Society, Future Leaders African Independent Researchers (FLAIR) Scheme (FLR\R1\201314) to JG.

**Competing interests:** The authors have declared that no competing interests exist.

## Results

The *T. vaginalis*-IMRS primers offered a higher test sensitivity of 0.03 fg/$\mu$L compared to the 18S rRNA PCR (0.714 pg/$\mu$L). The limit of detection for the Isothermal assay was 0.58 genome copies/mL. Using real-time PCR, the analytical sensitivity of the *T. vaginalis*-IMRS primers was <0.01 pg/$\mu$L, equivalent to less than one genome copy/$\mu$L.

## Conclusion

*De novo* genome mining of *T. vaginalis* IMRS as amplification primers serves as a platform for developing ultrasensitive diagnostics for Trichomoniasis and a wide range of infectious pathogens.

## Introduction

*Trichomonas vaginalis* (*T. vaginalis*) is considered the most common non-viral sexually transmitted infection (STI) [1]. So far, it affects approximately 174 million people globally, with more than half of these cases occurring in resource-limited regions, particularly in Africa, where access to testing laboratories is limited [1]. Most women infected with *T. vaginalis* present with vaginitis and vaginal discharge [2]. Untreated *T. vaginalis* infections are commonly associated with complications such as pelvic inflammatory disease (PID), adverse pregnancy outcomes (e.g., premature rupture of membranes, preterm delivery, and low-birthweight infants), prolonged Human Papilloma Virus infection and increased risk of HIV infection, mainly because it increases the shedding of viral proteins in genital tracts [2].

Usually, *T. vaginalis* proliferation by binary fission occurs on the mucosal surface of the urogenital tract of both males and females; however, females are more susceptible to infection than males [3]. Also, it has been shown that male infertility, epididymitis, and prostatitis are the most severe complications associated with *T. vaginalis* infection in males [3]. Therefore, accurate laboratory diagnosis of *T. vaginalis* is imperative in its treatment and control strategies.

The laboratory diagnosis of *T. vaginalis* in the clinical settings is done using wet-mount microscopy. However, this approach is not practical for screening large populations [3]. Also, since microscopy requires high parasite density at diagnosis, low density infections may be missed. Therefore, molecular techniques such as rRNA-based nucleic acid amplification tests (NAATs) [4] or DNA polymerase chain reaction (PCR) and real-time PCR assays and transcription-mediated amplification-based Aptima® assay [5] and in-pouch culture are a suitable alternative [6]. PCR based techniques detect *T. vaginalis* via specific gene targets such as the TVK3/7 and 18S rRNA gene [7]. Immunochromatographic methods have also been used to detect *T. vaginalis* infection [8]. Molecular tests have the advantage of having a short turnaround time and allow the use of non-invasive specimens such as urine and vaginal swabs [8]. However, these diagnostic techniques have numerous disadvantages. These include, high infrastructure costs, time-consuming procedures and labor-intensive protocols, the need to use standardized reagents and consumables, multistep reactions, false negative and positive results, interstrain variation, inefficiency in asymptomatic men or women, low sensitivity and specificity, and the need to use high density levels of the parasite [9]. Also, in resource constrained setups, access to prompt *T. vaginalis* diagnosis is limited by inadequate laboratory capacity [9]. For this reason, the World Health Organization (WHO) has developed algorithms

for syndromic case management [9]. However, this can lead to unnecessary treatment in many cases [9]. Even for symptomatic patients, accurate molecular diagnostic tests can increase the specificity of the syndromic management [9]. A study in South Africa that used NAATs to screen for *T. vaginalis* infection reported 50% asymptomatic cases in pregnant women visiting antenatal clinics. This points to the need of novel molecular diagnostics to detect asymptomatic *T. vaginalis* infections [10].

In this study, we demonstrate the unique ability of a highly sensitive molecular assay based on *de novo* genome mining strategy based on identical multiple repeat sequences (IMRS) [11] in the *T. vaginalis* genome to generate a primer set targeting numerous sequences. The primers are then used for PCR amplification, this guarantees improved sensitivity and specificity for *T. vaginalis* detection.

## Materials and methods

### Mining genomes using the IMRS algorithm

Primers were designed based on Identical Multi-Repeat Sequence (IMRS) genome mining algorithm as previously described [11] between 03/11/2020 and 10/02/2021. The IMRS algorithm is developed by adapting the Java Collection Framework by plugging in the Google Guava software (available at https://github.com/google/guava). The algorithm performs *ab initio* analysis of the annotated *T. vaginalis* genome to identify identical repeating oligonucleotide sequences of any given length.

The algorithm fragments the entire genome sequence into overlapping windows of size 'L' and enumerates all fragmented L-mer sequences into positional coordinates on the genome. The repeated L-mers are counted with their positions grouped and sorted based on the repeat count.

The hits are screened by computing positional coordinates for pair of repeat sequences that are adjacent to each other on the genome and within an amplifiable region, so that they can serve as a primer pair in amplification reactions. The specificity of lead pairs was evaluated by NIH's Basic Local Alignment Search Tool (BLAST) and NCBI Primer-BLAST, and the best pair was selected. The selected pair was capable of amplifying sequences from various locations of the *T. vaginalis* genome to generate amplicons of various lengths. Scaffolds and contigs were used to identify 69 repeats with expected product sizes of 76, 197, 318 and 439 bp on the *T. vaginalis* genome.

### *T. vaginalis* genomic DNA preparation

Quantitative Genomic DNA from *T. vaginalis* (ATCC® 30001DQTM) was obtained from the American Type Culture Collection (ATCC) at a concentration of $\geq$1 x $10^5$ copies/μL. *T. vaginalis* genomic DNA was diluted in Tris-EDTA buffer (Thermo Fisher Scientific, Waltham, Massachusetts, USA) to concentrations between $5.8{\times}10^2$ and $5.8{\times}10^{-4}$ genome copies/$\mu$L and used as template for both IMRS and 18S rRNA PCR assays.

### 18S rRNA PCR assay

The 18S rRNA PCR and IMRS assays were carried out in a reaction mixture containing dNTPs (Thermo Fisher Scientific, Waltham, Massachusetts, USA) (0.2 mM), forward and reverse primers (0.01 mM each), Taq Hot-Start DNA polymerase (Thermo Fisher Scientific, Waltham, Massachusetts, USA) (1.25 U), genomic template DNA (1 μL to a final PCR reaction volume of 25 μL). The cycling parameters were as follows: 95˚C for 3 min; 35 cycles of 95˚C for 30 s, 68˚C for 30 s, 72˚C for 30 s and 72˚C for 30 s followed by a final hold of 4˚C. The

negative controls consisted of all the PCR reaction components except template DNA, which was substituted with molecular grade water (Thermo Fisher Scientific, Waltham, Massachusetts, USA).

All PCR products were resolved in 2% agarose gel visualized on a UV Gel illuminator system (Fison Instruments, Glasgow, United Kingdom) under ethidium bromide staining. The negative controls consisted of all the PCR reaction components except template DNA which was substituted with molecular grade water (Thermo Fisher Scientific, Waltham, Massachusetts, USA).

## Isothermal IMRS amplification assay

Isothermal (Iso) IMRS amplification assays were performed in a 25 μL reaction mixture consisting of *Bst* 2.0 polymerase (640 U/mL) (New England Biolabs, Massachusetts, USA), with 1× isothermal amplification buffer, 3.2 μM forward primer, and 1.6 μM reverse primer (Jigsaw Bio solutions, Bengaluru, India) combined with 10 mM dNTPs (Thermo Fisher Scientific, Massachusetts, USA), 0.4 M Betaine (Sigma-Aldrich, Missouri, USA), molecular-grade water and Ficoll (0.4 g/mL) (Sigma-Aldrich, Missouri, USA) and genomic template DNA of 1 μL. Amplification was carried out at 56˚C for 40 min. Amplified products were visualized by gel electrophoresis in a 2% gel. The negative controls consisted of all the PCR reaction components except template DNA which was substituted with molecular grade water (Thermo Fisher Scientific, Waltham, Massachusetts, USA).

## Lower limit of detection

To assess the lower limit of detection (LLOD) of the *T. vaginalis* IMRS PCR assay, genomic DNA was diluted 100-fold from 100 pg/μL ($5.8\times10^2$ copies/μL) to $10^{-6}$ pg/μL (<1 copies/μL) and 10-fold from 100 pg/μL ($5.8\times10^2$ copies/μL) to $10^{-2}$ pg/μL (< 1 copies/μL) for the gold standard 18S rRNA PCR assay. Thereafter, 5 replicates of each dilution were used for the assays. Amplification products were visualized on gel after electrophoresis. To determine the LLOD of the *T. vaginalis* 18S rRNA and IMRS-PCR assays, probit analysis was performed using the ratio of successful reactions to the total number of reactions performed for each assay.

## Real-time PCR assay

The Quant Studio 5 Real-Time PCR System, with Quant Studio Design and Analysis Desktop Software v1.5., was used as a reference method for the *T. vaginalis* 18S rRNA PCR and the *T. vaginalis* IMRS-PCR assays as well as to determine the sensitivity of the *T. vaginalis* IMRS and 18S rRNA PCR primers for detecting *T. vaginalis* DNA. The genomic DNA was serially diluted 10-fold starting concentration of $10^4$ genome copies/μL. The final real-time PCR master mix volume was run 10 μL in triplicate and consisted of the following components: 5 μL SYBR Green qPCR Master Mix (Thermo Fisher, Massachusetts, USA), 1 μL forward and reverse IMRS primer mix (0.01 mM) 2.5 μL template genomic DNA and 1.5 μL molecular grade water. The amplification cycling conditions were 50˚C for 2 min; 95˚C for 10 min; 40 cycles of 95˚C for 15 s and 60˚C for 30 s.

## Cloning and characterization of *T. vaginalis* -IMRS amplicons

Gene cloning was performed to confirm the sequences of the amplicons obtained from the *T. vaginalis* IMRS PCR assay. *T. vaginalis* gDNA was amplified using Assembly IMRS-F (***TTCCGGATGGCTCGAGTTTTTCAGCAAGAT*** **GCTATATCTCATGATCTTAC**) and Assembly

IMRS-R (**_AGAATATTGTAGGAGATCTTCTAGAAAGAT_**<u>ACTATTTCCCTGCCGTTGGTGT</u>
<u>ATGTGCCGGATACCATTGTGTCA</u>) primers. The underlined bold sequences correspond to
the IMRS primers for amplifying the *T. vaginalis* genome, whereas the bold italicised corresponds to sequences in the cloning vector. The resulting amplicons were resolved on 2% agarose to confirm the fragment size and purified using the PureLink™ PCR purification kit
(Thermo Fisher). The purified amplicon was then ligated into pJET1.2/ blunt vector (Thermo
Fisher) using the NEBuilder® HiFi DNA Assembly kit (NEB) as per the manufacturer's
instructions. The resulting NEBuilder HiFi DNA Assembly product was transformed into
NEB 5-alpha Competent *E. coli* (NEB #C2987, NEB) following the manufacturer's instructions.
Transformed colonies were randomly selected, DNA extracted and Sanger-sequenced using
the universal pJET1.2 forward sequencing primer (CGACTCACTATAGGGAGAGCGGC) and
pJET1.2 reverse sequencing primer (AAGAACATCGATTTTCCATGGCAG). The resulting
nucleotides were trimmed and analysed using SnapGene software (GSL Biotech; available at
snapgene.com), aligned to check for similarity or "clonal" differences. BLAST was used to
check for similarity with the *T. vaginalis* genome.

## Sensitivity and specificity

The sensitivity of IMRS primers was tested using 17 *T. vaginalis*–PCR confirmed high vaginal
swab negative samples. Written informed consent was obtained from patients attending routine ante natal care enrolled between 08/03/2021 and 31/05/2021. The *in silico* analysis of specificity was checked in various ways. First, 76bp 197bp, 318bp and 439bp target sequences were
searched against the nucleotide database, with the BLAST-search tool being limited to exclude
*Trichomonas* spp. Primers for IMRS PCR were used to conduct *in silico* PCR using the NCBI
primer-BLAST tool and the *in silico* PCR tool in the UCSC genome browser on 12/04/2024.
The minimum perfect match for primers was set to a minimum of 10 perfect nucleotide
matches, and the amplification target was set to 4000 bp for *in silico* PCR using the *in silico*
PCR tool of the UCSC genome browser, while up to 6 mismatches and amplification of up to
4000 bp were allowed for *in silico* PCR using primer-BLAST.

## Copy number of amplification targets

The relationship between the genome size, DNA concentration, and the number of amplification targets was assessed using the mathematical formula for calculating dsDNA copy number
(https://www.technologynetworks.com). The genome size for *T. vaginalis* was obtained from
literature, while the genomic DNA concentration was provided by the vendor.

## Data analysis

Graphs were plotted with GraphPad Prism version 7.0 (GraphPad Software, San Diego, CA,
USA). The mean, and SD values were calculated in Excel 2016. To determine the LLOD of *T.
vaginalis* -IMRS and *T. vaginalis* -18S rRNA PCR assays (the concentration at which genomic
*T. vaginalis* DNA is detected with 95% confidence), probit regression analyses were performed
in Excel 2016. Statistical analyses were performed using a *t*-test of GraphPad Prism version
7.0. For two-tailed distributions, $P < 0.05$ was considered significant.

## Ethical consideration

This study was reviewed and approved by the Mount Kenya University Ethical Review Committee under reference MKU/ERC/1649.

## Results

### Location of IMRS primer targets on the *T. vaginalis* genome

Repeat sequences that could be used as forward and reverse primers for an amplification assay using the IMRS based genome mining algorithm were identified (10). A total of 69 repeat sequences of 76bp, 197bp, 318bp and 439bp were identified, and the targeted regions are shown in Fig 1. These repeats can inter-changeably serve as forward or reverse primers due to their presence in opposite orientations at various loci of the sense and antisense strands as depicted using a circos plot (Fig 1). It was hypothesized that the identified primer pair F 5'- GCTATATCTCATGATCTTAC -3' and R 5'- ATGTGCCGGATACCATTGTGTCA - 3' would generate many amplicons, leading to increased analytical sensitivity in PCR assays.

### Amplification of sequences on *T. vaginalis* genome

The sensitivity of the IMRS primers to amplify the targeted regions in the *T. vaginalis* genome was confirmed by serially diluting genomic DNA. Dilutions were then used as a template for PCR amplification. The IMRS primers could detect *T. vaginalis* genomic DNA from a concentration less than 1 fg/μL (S1 Fig). The gold standard 18S rRNA primers (S2 Fig) could also detect *T. vaginalis* genomic DNA down to a concentration of 1 fg/μL.

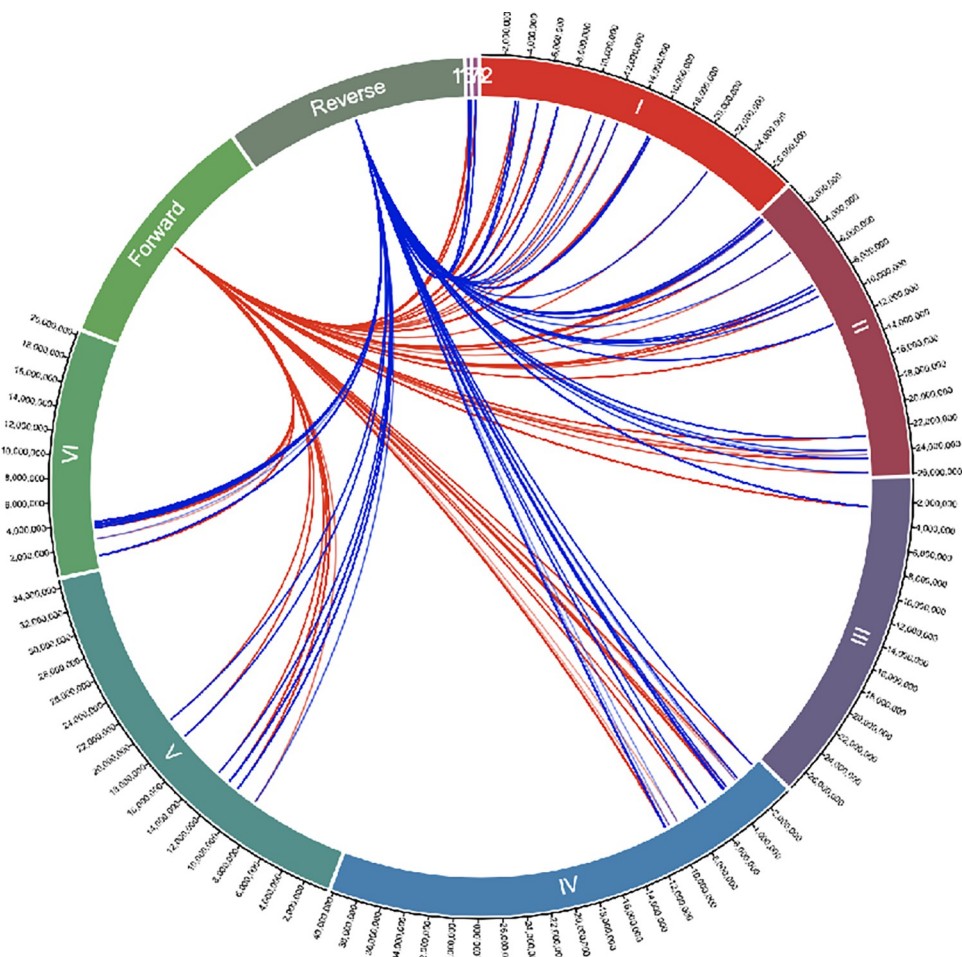

**Fig 1. Circos plot displaying the Identical Multi-Repeat Sequence (IMRS) primer target regions on the *Trichomonas vaginalis* genome.** As shown, the primers target a total of 69 repeat sequences on the genome. The red lines show the forward primer targets while the blue lines show the reverse primer targets.

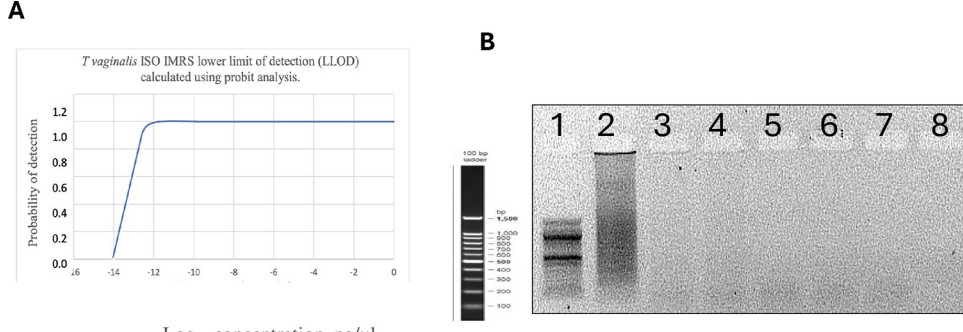

**Fig 2. Iso-thermal amplification of *Trichomonas vaginalis* DNA using TV-IMRS primers.** Lower limit of detection (LLOD) for the Iso-thermal amplification was calculated using the Probit analysis (A). 2% gel image of amplicons of 100-fold serial dilution of *Trichomonas vaginalis* DNA template (1, 100bp ladder, 2, $10^2$ pg/µl, 3, 1 pg/µl, 4, $10^{-2}$ pg/µl, 5, $10^{-4}$ pg/µl, 6, $10^{-6}$ pg/µl, 7, $10^{-8}$ pg/µl and 8, non-template control). The estimated LLOD for the Isothermal assay was 0.0201 pg/µl.

## Isothermal amplification of genomic *T. vaginalis* DNA

Serially diluted genomic DNA was used to perform Isothermal *T. vaginalis* -IMRS amplification. The LLOD for the *T. vaginalis* -Iso-IMRS assay was estimated at 0.0201 pg/µl (Fig 2A). As shown in Fig 2B, the reaction products were visualized on a 2% gel. The Iso- *T. vaginalis* -IMRS assay successfully amplified *T. vaginalis* DNA at a concentration of $5.84 \times 10^2$ copies/µL.

## Gene sequencing of *T. vaginalis* assembly products

To confirm the exact regions amplified by the *T. vaginalis* -IMRS primers, we cloned the amplicon into blunt cloning vectors and transformed it into electrocompetent *E. coli* cells that were then plated onto agar plates. Assembly products were then separated on 2% gel (Fig 3). DNA from transformed *E. coli* cells was extracted and sequenced. Multiple sequencing alignment confirmed *T. vaginalis* sequences. These results suggested that the *T. vaginalis* -IMRS primers were specific for targets within the genome (Fig 4).

## Lower Limit of Detection (LLOD) calculation

To determine the lowest limit of detection (LLOD) of the IMRS PCR assay relative to the gold standard 18S rRNA PCR, probit statistic was performed using *T. vaginalis* genomic DNA serially diluted 100-fold and 10-fold and used as a template for the *T. vaginalis* -IMRS and 18S RNA PCR assays, respectively (Table 1). Fig 5A shows the probit plot for the *T. vaginalis* -IMRS PCR assay, and Fig 5B shows the probit plot for the gold standard 18S rRNA PCR assay. The LLOD was calculated as the concentration at which *T. vaginalis* DNA can be detected with 95% confidence. As indicated, the IMRS primers for *T. vaginalis* had an LLOD = 0.03 fg/µL, Fig 5A. The gold standard primers for *T. vaginalis* had an LLOD = 0.714 pg/µL, 5B. The probit analysis showed that the *T. vaginalis* -IMRS PCR assay had increased sensitivity compared to the gold standard 18S rRNA PCR assay.

## Minimum concentration detection of *T. vaginalis* genomic DNA

Real-time PCR assay was also performed using serially diluted genomic *T. vaginalis* DNA as a template and *T. vaginalis* -IMRS primers or *T. vaginalis* -18S rRNA primers. The mean Ct values at each dilution were used to plot amplification bar graphs (Fig 6 for *T. vaginalis* -IMRS

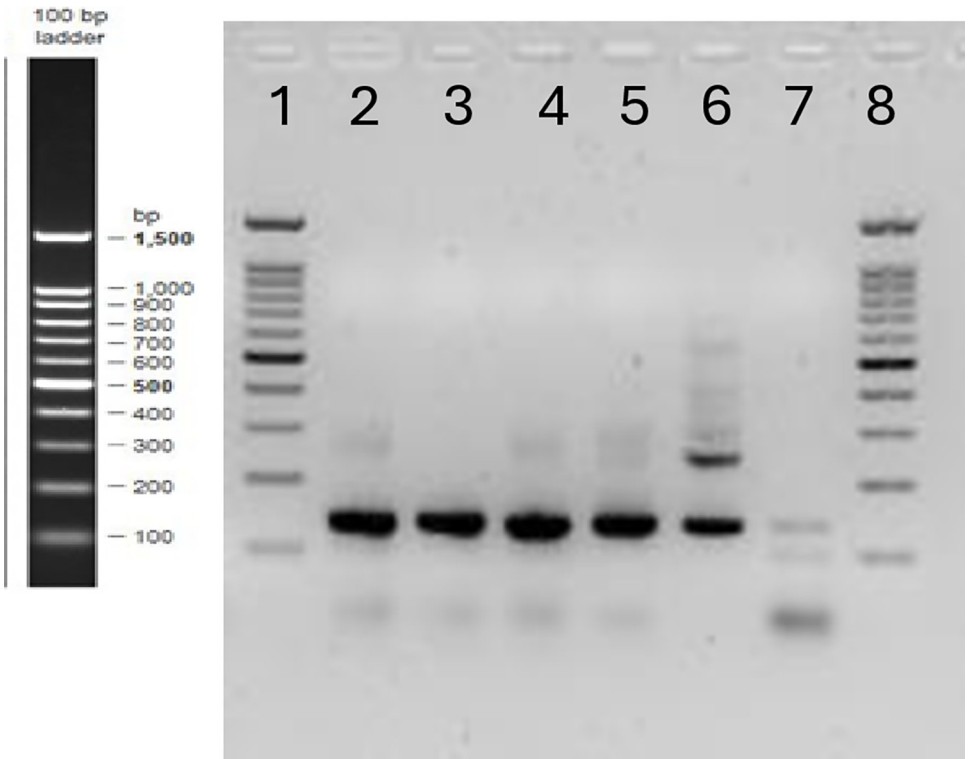

**Fig 3. 2% gel image showing gene cloning assembly products.** As shown, a total of five *E. coli* competent cells (No. 2–6) were transformed and DNA extracted and sequenced.

primers and for *T. vaginalis* -18S rRNA primers). The *T. vaginalis* -IMRS primers detected genomic DNA to a concentration <0.01 pg/µL, equivalent to less than one genome copy/µL.

## Specificity of the *T. vaginalis* -IMRS primers

The BLAST results indicated no significant similarity between nucleotide sequences on the database, with 76bp, 197bp, 318bp, and 439bp, except for *T. vaginalis*. Further BLAST analysis with other STI genomes indicated that the sequences were not similar to *Chlamydia trachomatis (*taxid: 813*), Treponema pallidum* (taxid: 160) and *Neisseria gonorrhoea* (taxid: 485). Also, the *T. vaginalis* -IMRS primers were non-specific to PCR negative samples (S3 Fig).

## Discussion

One of the major challenges in eradicating *T. vaginalis* is the accurate identification of low-density infections. Therefore, highly sensitive, efficient, and reliable molecular diagnostic techniques are urgently needed [12–14]. Once developed, these diagnostic assays will be vital in detecting asymptomatic *T. vaginalis* cases, therefore controlling the emergence and spread of infections. This is a key priority action for the World Health Organization (WHO) in strengthening efforts to ensure high-quality diagnostics assays for STIs are accessible and available particularly in resource limited countries [15].

This study investigated the sensitivity of the novel *T. vaginalis* -IMRS PCR assay for detecting genomic *T. vaginalis* DNA. Compared to the conventional 18S-rRNA PCR assay, the *T. vaginalis* -IMRS PCR assay detected serially diluted genomic DNA up to a concentration of <1 fg/µL. This demonstrates the potential applicability of the *T. vaginalis* -IMRS PCR assay in

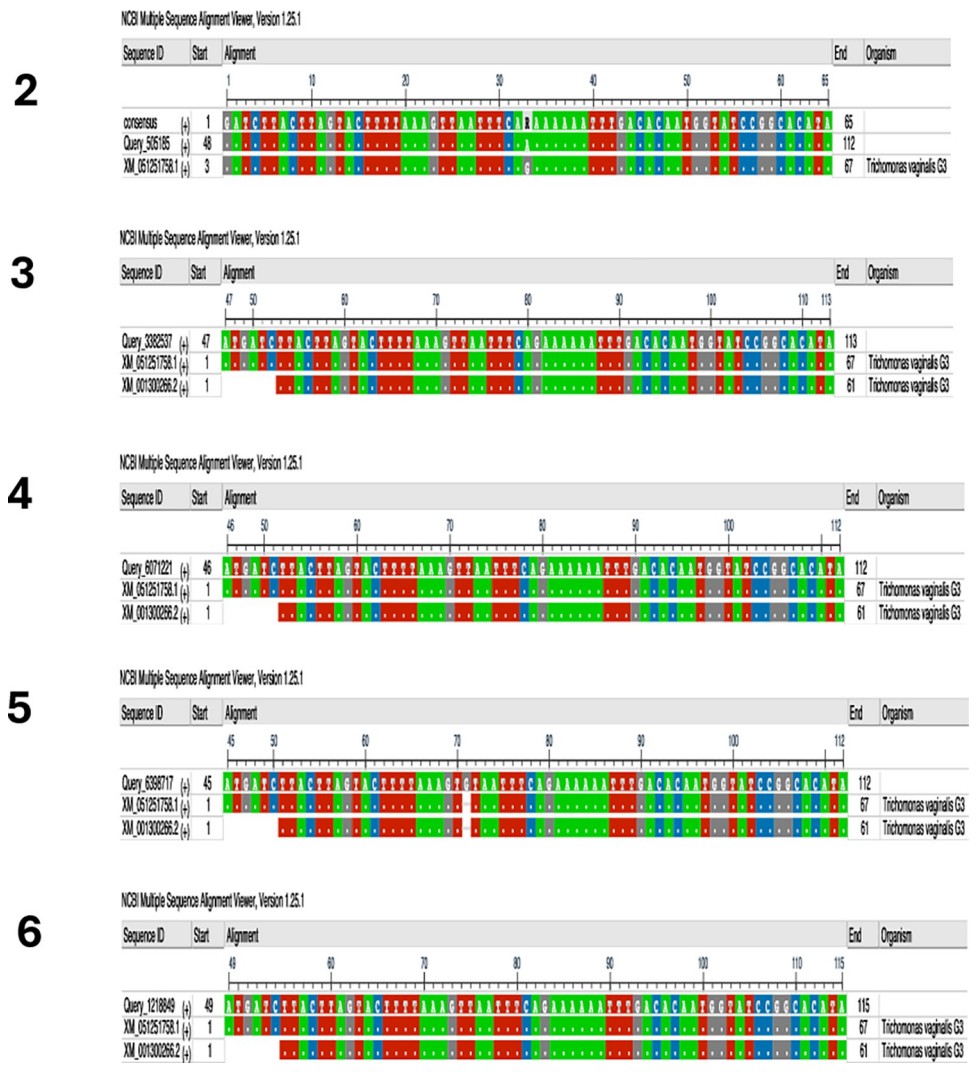

**Fig 4. Multiple sequencing alignment analysis of five assembly products obtained in Fig 3 using the *Trichomonas vaginalis* IMRS primers.** The numbers correspond to the gel image assembly products in Fig 3.

detecting asymptomatic clinical infections. Indeed, numerous studies have reported the occurrence of asymptomatic STI cases [16–18], and routine testing is therefore critical to mitigate their spread and potentially slow the development of antibiotic resistance [19]. Although culture is considered the gold standard technique in diagnosing *T. vaginalis*, approximately 300 organisms are required for a positive test [20, 21]. For low density infections, the use of PCR is usually recommended. A previous study reported an analytical sensitivity of 10 fg and a detection limit of one whole flagellated cell per 25 µL of PCR mixture using the 18S rDNA PCR assay [20–22].

A study that compared the performance characteristics of microscopy, culture, point-of-care tests, Aptima and Real-time PCR methods for detecting *T. vaginalis* infection from vaginal swab samples reported poor sensitivity (38%) using microscopy [23]. These tests that identified *T. vaginalis* in symptomatic women had an average specificity of >98% [24]. In another study, the sensitivity of the Aptima and Real-time PCR was 100% while that of microscopy was 81.8% [25]. Our research, however, reported a higher analytical sensitivity of the *T. vaginalis*

**Table 1. Dilution of DNA template to determine the lower limit of detection of PCR reactions.**

| IMRS | | 18S rRNA | |
|---|---|---|---|
| Serial dilutions (pg/µl) | Replicates (5) | Serial dilutions (pg/µl) | Replicates (5) |
| 100 | 5/5 | 100 | 5/5 |
| 1 | 5/5 | 10 | 4/5 |
| 0.01 | 5/5 | 1 | 3/5 |
| 0.0001 | 5/5 | 0.1 | 2/5 |
| 0.000001 | 5/5 | 0.01 | 2/5 |
| 0.00000001 | 5/5 | 0.001 | 2/5 |
| 1E-10 | 0/5 | 0.0001 | 1/5 |
| Coefficient | -0.4123 | Coefficient | -7.7857 |
| *P-Value | 0.9994 | P-Value | 0.584 |

IMRS–Identical Multi-Repeat Sequence, 18S rRNA– 18S ribosomal Ribonucleic Acid

*P-Values were estimated using probit analysis

-IMRS PCR assay. This, therefore, demonstrates the increased sensitivity of the novel assay in the detection of *T. vaginalis* infection. We also determined the minimum concentration at which the *T. vaginalis* IMRS primers could detect genomic DNA using real-time PCR. Our research reported a Ct value of 32 (Fig 5). This finding is consistent with a previous study that investigated the clinical performance of BD CTGCTV2 (CTGCTV2) assay on the BD COR System (COR) for the detection of *T. vaginalis* that reported a Ct value of 31 [26]. However, another study reported a weak positive Ct value of >37 in patients who had completed treatment [27]. This highlights the possibility of real-time PCR reporting clinically irrelevant positives [27]. This is usually common with patients with a slow parasite clearance rate where DNA from dead trichomonads could influence the diagnosis of *T. vaginalis* using real-time PCR [27, 28].

A novel NAAT called loop-mediated isothermal amplification (LAMP) [29] has been established). This assay uses a DNA polymerase with strand displacement activity and a set of four

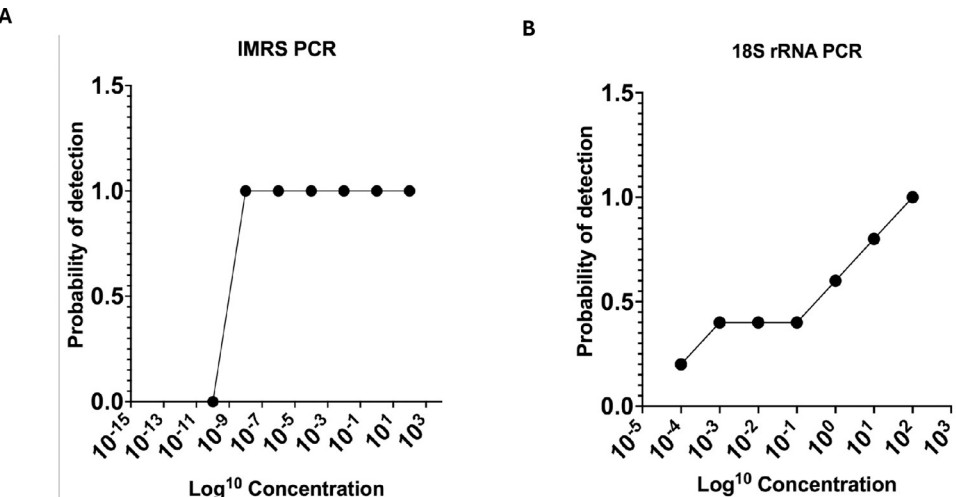

**Fig 5. Probit regression analysis to calculate the lower limit of detection for the *Trichomonas vaginalis* -IMRS primers and the 18S rRNA PCR assay.** Lower limit of detection for *Trichomonas vaginalis* using IMRS primers was 0.03 fg/µl (A) and 18S rRNA PCR primers was 0.714pg/µL (B) respectively.

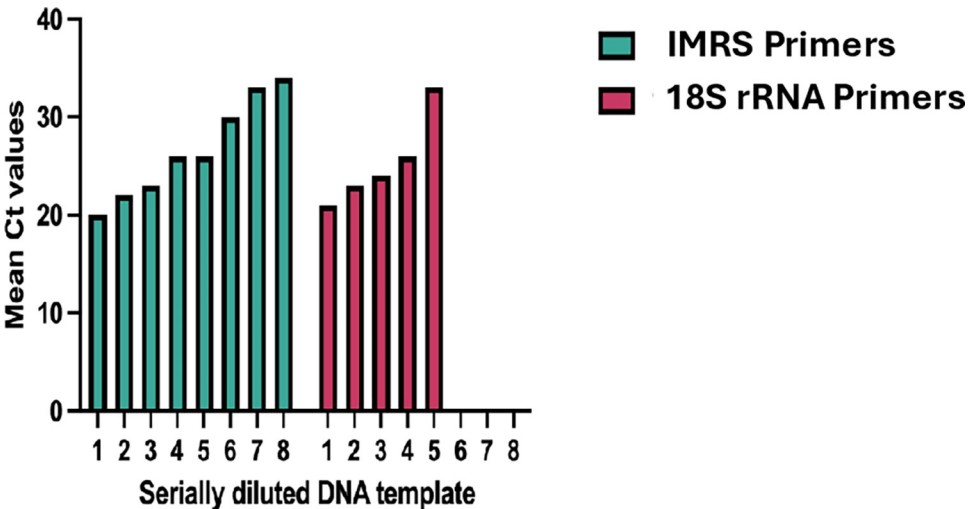

**Fig 6. Mean Ct values using 10-fold serially diluted genomic *Trichomonas vaginalis* DNA for the IMRS and 18S rRNA RT-PCR assay, respectively.** In total, 8 dilutions were done, and each dilution served as template for the RT-PCR assay. The DNA concentration of the starting DNA template was 100 pg/μL.

to six primers that amplify nucleic acids under isothermal conditions between 60–65°C; therefore, incubation can be done in a heat block or water bath [30, 31]. This has the advantage of easily being deployed in the field. To date, the application of LAMP assays has been developed for laboratory diagnosis of infectious diseases [24, 32]. The *T. vaginalis* iso-thermal shares the same principle as the LAMP assay except for the number of primer sets used to amplify the DNA material. We reported an analytical sensitivity of $5.82 \times 10^2$ copies/μl. This finding contrasts with what other studies have confirmed. For instance, a detection limit of $10^0$ trichomonads/mL or 1 trichomonad for both spiked genital swab and urine specimens was reported [33]. These discrepancies could be attributed to sample preparation procedures.

This study has some limitations. First, access to vaginal *T. vaginalis* samples to compare the IMRS tests was limited. Also, the *in silico* PCR to confirm the analytical specificity was only limited by the genomic data available at primer-BLAST and UCSC genome browser. However, to the best level of our understanding, the study reports the first rapid and accurate IMRS real-time PCR for *T. vaginalis*, and the findings presented might positively impact the future development of sensitive assays for detecting parasites, particularly *T. vaginalis*, whose genomes are large, and have many repeating sequences.

## Conclusion

The *T. vaginalis* -IMRS isothermal amplification assay removes the numerous challenges other NAAT assays face. Its performance may be improved by using fluorescent tags to achieve a visual read out signal and used reliably for the detection of *T. vaginalis* in a simple, sensitive assay format, providing an alternative to more complex molecular tests for diagnosis. Additionally, the assay eliminates challenges associated with conventional molecular tests and provides new opportunities to diagnose *T. vaginalis* in point of care settings. To accurately interpret amplicons from IMRS PCR assays, qPCR using specific probes can be optimized.

## Supporting information

**S1 Fig. Image of 10-fold serially diluted (1 and 28, 100bp ladder 2–100, 3–10, 4–1, 5–0.1, 6–0.01, 7–0.01, 8–0.001, 9–0.0001, 10–$10^{-4}$, 11–$10^{-5}$, 12, - $10^{-6}$, 13–$10^{-7}$, 14–$10^{-8}$, 15–$10^{-9}$,**

**16–$10^{-10}$, 17–$10^{-11}$, 18–$10^{-12}$, 19–$10^{-13}$, 20–$10^{-14}$, 21–$10^{-15}$, 22–$10^{-16}$, 23–$10^{-17}$, 24–$10^{-18}$, 25 -,$10^{-19}$ 26-$10^{-20}$ and 27 -Non Template Control (NTC) (pg/µl) genomic *Trichomonas vaginalis* DNA amplicons resolved on 2% gel using IMRS primers.**
(TIF)

**S2 Fig. Image of 10-fold serially diluted (1, 100bp ladder 2–100, 3–10, 4–1, 5–0.1, 6–0.01, 7–0.01, 8–0.001, 9–0.0001, 10 –Non Template Control (NTC) (pg/µl) genomic *Trichomonas vaginalis* DNA amplicons resolved on 2% gel using gold standard 18S rRNA PCR primers.**
(TIF)

**S3 Fig. 2% gel image of PCR confirmed *Trichomonas vaginalis* negative samples numbers 2–10, and 12–19 and 20 Non Template Control (NTC) (pg/µl).** Well number 1 and 11 is 100bp ladder.
(TIFF)

## Author Contributions

**Conceptualization:** Clement Shiluli, Shwetha Kamath, Racheal Kimani, Hussein M. Abkallo, Michael Maina, Harrison Waweru, Moses Kamita, Nicole Pamme, Catherine M. Klapperich, Srinivasa Raju Lolabattu, Jesse Gitaka.

**Data curation:** Clement Shiluli, Shwetha Kamath, Bernard Oduor, Hussein M. Abkallo, Michael Maina, Moses Kamita, Nicole Pamme, Joshua Dupaty, Catherine M. Klapperich, Srinivasa Raju Lolabattu, Jesse Gitaka.

**Formal analysis:** Clement Shiluli, Shwetha Kamath, Bernard N. Kanoi, Racheal Kimani, Srinivasa Raju Lolabattu, Jesse Gitaka.

**Funding acquisition:** Nicole Pamme, Catherine M. Klapperich, Jesse Gitaka.

**Investigation:** Bernard N. Kanoi, Bernard Oduor, Michael Maina, Harrison Waweru, Moses Kamita, Catherine M. Klapperich, Srinivasa Raju Lolabattu.

**Methodology:** Clement Shiluli, Bernard N. Kanoi, Bernard Oduor, Michael Maina, Nicole Pamme, Joshua Dupaty, Srinivasa Raju Lolabattu, Jesse Gitaka.

**Project administration:** Racheal Kimani, Catherine M. Klapperich.

**Resources:** Shwetha Kamath, Racheal Kimani, Hussein M. Abkallo, Moses Kamita, Nicole Pamme, Catherine M. Klapperich, Srinivasa Raju Lolabattu, Jesse Gitaka.

**Software:** Clement Shiluli, Bernard Oduor, Michael Maina, Nicole Pamme, Catherine M. Klapperich, Srinivasa Raju Lolabattu, Jesse Gitaka.

**Supervision:** Bernard N. Kanoi, Racheal Kimani, Harrison Waweru, Moses Kamita, Jesse Gitaka.

**Validation:** Clement Shiluli, Bernard Oduor, Moses Kamita, Nicole Pamme, Joshua Dupaty, Catherine M. Klapperich, Srinivasa Raju Lolabattu.

**Visualization:** Clement Shiluli, Michael Maina, Moses Kamita, Nicole Pamme, Joshua Dupaty, Srinivasa Raju Lolabattu, Jesse Gitaka.

**Writing – original draft:** Clement Shiluli, Shwetha Kamath, Bernard N. Kanoi, Racheal Kimani, Bernard Oduor, Hussein M. Abkallo, Michael Maina, Harrison Waweru, Moses

Kamita, Nicole Pamme, Joshua Dupaty, Catherine M. Klapperich, Srinivasa Raju Lolabattu, Jesse Gitaka.

**Writing – review & editing:** Clement Shiluli, Bernard N. Kanoi, Racheal Kimani, Hussein M. Abkallo, Michael Maina, Harrison Waweru, Moses Kamita, Nicole Pamme, Joshua Dupaty, Catherine M. Klapperich, Srinivasa Raju Lolabattu, Jesse Gitaka.

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
