## [Decision Letter · Decision Letter 0]

19 Nov 2024

PONE-D-24-18894Highly Sensitive Molecular Assay Based on Identical Multi-Repeat Sequence (IMRS) Algorithm for the Detection of Trichomonas vaginalis InfectionPLOS ONE

Dear Dr. Gitaka,

Thank you for submitting your manuscript to PLOS ONE. After careful consideration, we feel that it has merit but does not fully meet PLOS ONE’s publication criteria as it currently stands. Therefore, we invite you to submit a revised version of the manuscript that addresses the points raised during the review process.

We look forward to receiving your revised manuscript.

Kind regards,

Adriana Calderaro

Academic Editor

PLOS ONE

“This research was supported by the Royal Society, Future Leaders African Independent Researchers (FLAIR) Scheme (FLR\\R1\\201314) to JG”

6. Please upload a copy of Supporting Information Figures 1-3 which you refer to in your text on page 20.

7. PLOS ONE now requires that authors provide the original uncropped and unadjusted images underlying all blot or gel results reported in a submission’s figures or Supporting Information files. This policy and the journal’s other requirements for blot/gel reporting and figure preparation are described in detail at https://journals.plos.org/plosone/s/figures#loc-blot-and-gel-reporting-requirements and https://journals.plos.org/plosone/s/figures#loc-preparing-figures-from-image-files. When you submit your revised manuscript, please ensure that your figures adhere fully to these guidelines and provide the original underlying images for all blot or gel data reported in your submission. See the following link for instructions on providing the original image data: https://journals.plos.org/plosone/s/figures#loc-original-images-for-blots-and-gels.  

Reviewers' comments:

Reviewer's Responses to Questions

**Comments to the Author**

1. Is the manuscript technically sound, and do the data support the conclusions?

Reviewer #1: Yes

2. Has the statistical analysis been performed appropriately and rigorously? 

Reviewer #1: Yes

3. Have the authors made all data underlying the findings in their manuscript fully available?

Reviewer #1: Yes

4. Is the manuscript presented in an intelligible fashion and written in standard English?

Reviewer #1: Yes

5. Review Comments to the Author

Reviewer #1: The manuscript describes a sensitive isothermal amplification technique targeting repetitive sequences on an important pathogen. However, it should have been tested on more clinical samples including vaginal swabs, genital secretions, urine and should also be evaluated against microscopy and culture. Discussion should include few more studies especially those demonstrating a good sensitivity and specificity. Since authors from a commercial entity , the conflict of interest statement should clearly mention the same.

6. PLOS authors have the option to publish the peer review history of their article (what does this mean?). If published, this will include your full peer review and any attached files.

Reviewer #1: No

---

## [Author Response · Author response to Decision Letter 0]

2 Jan 2025

Dear Adriana Calderaro

2nd January 2025.

Ref: Submission of revised research article

On behalf of the authors, I extend my gratitude to the reviewer on the substantial input on our submitted manuscript. Indeed, the revised manuscript has been greatly improved. 

We have addressed the comments recommended by the reviewer and academic editor as indicated below;

1. However, it should have been tested on more clinical samples including vaginal swabs, genital secretions, urine and should also be evaluated against microscopy and culture. 

We are in agreement with the above comment. Our intention was to validate the assay with more clinical samples. We only had access to 17 clinical samples that were used for assay validation. We have acknowledged that this was a limitation to our study (Line 389) and recommended that further validation with more samples is needed. However, to the best level of our knowledge, our study reports a rapid and accurate IMRS PCR assay for T. vaginalis, and the findings presented might positively impact the future development of sensitive assays for detecting parasites T. vaginalis. 

2. Discussion should include few more studies especially those demonstrating a good sensitivity and specificity. 

Thanks for the comment. We have included two research articles (reference 23 and 24) that highlights the performance characteristics of five different T. vaginalis diagnostic methods (Line 350 - 355).

3. Since authors from a commercial entity, the conflict of interest statement should clearly mention the same.

Thanks for the suggestion. Our conflict-of-interest statement has been revised accordingly to include authors affiliated to the commercial company (Line 420).

4. Ethics statement.

The ethics statement only appears in the methodology section of the manuscript. Line 264.

5. Financial disclosure statement

Thanks for the suggestion. The financial statement has been revised as suggested (Line 427 - 429).

6. Supporting figures.

Supporting Figures 1 – 3 have been uploaded as suggested. Raw Gel images have also been uploaded and are available on this link https://doi.org/10.5281/zenodo.14589291.

Sincerely,

Clement Shiluli, 

Mount Kenya University, Main Campus

Thika, Kenya

---

## [Editor Report · Decision Letter 1]

7 Jan 2025

Highly Sensitive Molecular Assay Based on Identical Multi-Repeat Sequence (IMRS) Algorithm for the Detection of Trichomonas vaginalis Infection

PONE-D-24-18894R1

Dear Dr. Gitaka,

We’re pleased to inform you that your manuscript has been judged scientifically suitable for publication and will be formally accepted for publication once it meets all outstanding technical requirements.

Kind regards,

Adriana Calderaro

Academic Editor

PLOS ONE

Additional Editor Comments (optional):

Thanks to have responded to the reviewers' comments.
---

## [Editor Report · Acceptance letter]

27 Jan 2025

PONE-D-24-18894R1 

PLOS ONE

Dear Dr. Gitaka, 

I'm pleased to inform you that your manuscript has been deemed suitable for publication in PLOS ONE. Congratulations! Your manuscript is now being handed over to our production team.

Kind regards, 

on behalf of

MD, PhD, Full Professor Adriana Calderaro 

Academic Editor

PLOS ONE